# The Effect of Ursodeoxycholic Acid on Small Intestinal Bacterial Overgrowth in Patients with Functional Dyspepsia: A Pilot Randomized Controlled Trial

**DOI:** 10.3390/nu12051410

**Published:** 2020-05-14

**Authors:** Bom-Taeck Kim, Kwang-Min Kim, Kyu-Nam Kim

**Affiliations:** Department of Family Practice and Community Health, Ajou University School of Medicine, Suwon 16499, Korea; lovesong@ajou.ac.kr (B.-T.K.); gaksi@ajou.ac.kr (K.-M.K.)

**Keywords:** functional dyspepsia, small intestinal bacterial overgrowth, ursodeoxycholic acid

## Abstract

Functional dyspepsia (FD) is associated with small intestinal bacterial overgrowth (SIBO). Several animal studies have reported that ursodeoxycholic acid (UDCA) has antibacterial and anti-inflammatory effects in the intestine. We hypothesized that UDCA may be effective against dyspeptic symptoms and SIBO in patients with FD. We conducted this randomized controlled trial to investigate the effects of UDCA in FD patients with SIBO. Twenty-four patients diagnosed with FD and SIBO based on lactulose breath test (LBT) were randomly assigned to either a UDCA treatment group or an untreated group. The treatment group received 100 mg of UDCA three times per day for two months; the untreated group was monitored for two months without intervention. After two months in both groups, we reevaluated LBT and FD symptoms using the Nepean dyspepsia index-K. FD symptoms in the UDCA-treated group were significantly reduced after two months compared with baseline and FD symptom scores between the UDCA-treated and untreated groups showed statistically significant differences after two months. In addition, the total methane gas levels for 90 minutes in LBT were significantly decreased after two months compared with baseline in the UDCA-treated group. In this preliminary exploratory study, we found that two months of UDCA treatment resulted in FD symptom improvement and reduced methane values during 90 minutes on the LBT, suggesting that methane-producing SIBO were associated with symptoms of dyspepsia and that UDCA was helpful in these patients. These findings need to be validated via large-scale controlled and well-designed studies.

## 1. Introduction

Functional dyspepsia (FD) is a functional digestive disorder characterized by one or more of the following symptoms: postprandial fullness, early satiation, epigastric pain and epigastric burning [1]. Although the causes of FD include *Helicobacter pylori* infection, visceral hypersensitivity and disturbed gastric motility [2,3,4,5], emerging evidence suggests that intestinal bacterial imbalance, particularly small intestinal bacterial overgrowth (SIBO), may also be one of the causes underlying FD [6,7]. Indeed, a shift in paradigm suggests that the role of SIBO is important in the development of FD. In a recent study, SIBO measured by hydrogen breath test was indicated for the clinical management of patients with suspected FD [8]. In another study, FD patients’ symptoms improved after two weeks of treatment with non-absorbing antibiotic rifaximin, which was effective in treating SIBO [9].

Ursodeoxycholic acid (UDCA) is one of the secondary bile acids secreted in the body and is used as a representative liver supplement. Primary bile acids (cholic and chenodeoxycholic acids) formed in the liver are secreted into the intestine, and upon encountering beneficial bacteria in the intestine, form secondary bile acids such as deoxycholic acid, lithocholic acid and UDCA via dehydroxylation [10,11]. Secondary bile acids perform classical functions including digestion of dietary lipids and fat-soluble vitamins in the intestine; however, recent animal studies have reported that these acids, especially UDCA, play an important role in balancing intestinal bacteria by inhibiting harmful bacteria [12,13]. In addition, some studies have demonstrated the advantages of UDCA in patients with gut dysbiosis [14,15]. Thus, UDCA has direct or indirect antimicrobial effects and modulates the composition of microbiota [16]. However, these studies in humans have focused on the bacterial imbalance in the large intestine, and no study has investigated the effect of UDCA on FD combined with SIBO. Therefore, we hypothesize that treatment with UDCA improves upper gastrointestinal symptoms and SIBO in patients with FD. In this randomized controlled trial, we investigated whether dyspepsia symptoms and SIBO improved when FD patients whose SIBO was confirmed by a breath test were treated with UDCA.

## 2. Materials and Methods

### 2.1. Study Populations and Design

We recruited patients with FD corresponding to Rome IV via detailed interviews during their visit to Ajou University Health Promotion Center or Department of Family Medicine as outpatients with digestive symptoms from May 2019 to December 2019. The Rome IV criteria define FD as one of the four symptoms (postprandial fullness, early satiation, epigastric pain and epigastric burning) occurring for at least six months before diagnosis and persisting for the last three months. In addition, endoscopic examination should reveal no underlying structural abnormalities.

All patients diagnosed with FD underwent the hydrogen/methane breath test for SIBO and the urea breath test for *Helicobacter pylori* infection. Patients with FD were excluded from this study if they had a history of: (i) gastrointestinal ulcers or inflammatory bowel diseases such as Crohn’s disease or ulcerative colitis; (ii) gastrectomy or intestinal resection (except appendectomy); or (iii) viral liver disease (types A, B and C), cirrhosis or any cancer; (iv) blood tests whose serum aspartate aminotransferase (AST), alanine aminotransferase (ALT) or gamma-glutamyltransferase (GGT) levels were more than three-fold higher than the normal range or whose serum creatinine levels were greater than 1.5-fold the normal range;(v) chronic renal failure, congestive heart failure and thyroid disease; (vi) persistent alcohol consumption (at least four bottles of soju per week, i.e., at least 240 g of alcohol per week); or (vii) severe biliary obstruction.

We confirmed each patient’s medical history by esophagogastroduodenoscopy and blood tests performed at the health promotion center or on an outpatient basis within the three months prior to screening. Colonoscopy findings were based on patients’ medical records within the last two years. Twenty-four patients with SIBO diagnosed with FD were enrolled and randomly assigned (1:1) to either the UDCA-treated (URUSA^®^; Daewoong Pharmaceutical, Seoul, Korea) or the untreated group via block randomization. The UDCA group received 100 mg of UDCA three times per day for two months, and the untreated group was monitored for two months without intervention. Participants in both the UDCA and untreated groups completed a questionnaire for FD and a breath test for SIBO at baseline and two months later.

We planned the current study including 10 patients in the UDCA group and 10 in the untreated group and recruited 12 participants in each group considering the dropout rate of 20%. After two months, two subjects dropped out of the UDCA-treated group (one refused to participate in the study because of epigastric soreness and another withdrew consent), and three subjects from the untreated group withdrew consent. Finally, 10 patients in the UDCA-treated group and nine in the untreated group completed the study (Figure 1).

In both groups, participants were banned from using antibiotics, prebiotics and probiotics and medications (H2 blocker, proton pump inhibitor and prokinetics). We classified patients into smokers and nonsmokers depending on their smoking history at the time of hospital visit. We measured alcohol consumption as grams of ethanol consumed per week using the graduated frequency method [17]. We confirmed the presence or absence of current history of hypertension, diabetes and hyperlipidemia via medical records or interviews.

All subjects provided informed consent for inclusion before they participated in the study. The study was conducted in accordance with the Declaration of Helsinki, and the protocol was approved by the Ethics Committee of Ajou University Hospital (approval no.: AJIRB-MED-CT4-18–451). I have received Written Inform Consent from all participant patients. The clinical trial number was acquired from the Clinical Research Information Service (CRIS registration number: KCT0004910).

### 2.2. Measurements

#### 2.2.1. Gastrointestinal Symptoms

The questionnaire about dyspepsia measured each patient’s Nepean dyspepsia index-K (NDI-K), which has been validated in Korea [18]. The NDI-K is based on 15 symptoms: epigastric pain, epigastric discomfort, epigastric soreness, chest soreness, stomach cramps, chest pain, early satiation, gastric acid reflux, postprandial fullness, epigastric pressure, upper abdominal bloating, nausea, belching, vomiting and poor breathing. Frequency of symptoms was scored along five time ranges based on the number of days the patient manifested the symptoms over the previous two weeks (0 = not at all, 1 = 1–4 days, 2 = 5–8 days, 3 = 9–12 days, 4 = daily or almost every day). The intensity of the symptoms was measured as one of five levels: 0 = none, 1 = very weak, 2 = weak, 3 = slightly severe, 4 = severe and 5 = very severe. Because the normal range of FD using NDI-K has yet to be defined, we defined adequate relief of FD symptoms as NDI-K score reduction greater than 30% compared with the baseline.

#### 2.2.2. Lactulose Breath Test

The lactulose breath test (LBT) was performed after 12 hours of fasting for baseline measurements before the subjects ingested 15 mL of syrup containing 10 g lactulose (Duphalac®; Choongwae Pharma Corporation, Seoul, Korea). After patients consumed 10 g of the lactulose, their breath was tested every 20 minutes during the first hour and then every 15 minutes per hour, subsequently. The hydrogen/methane concentration in the sample was measured immediately using a Breath Tracker SC Quintron gas chromatograph (Quintron Instrument Company, Milwaukee, WI, USA). We defined a positive LBT based on one of the following criteria: 1) a CH_4_ ≥ 10 ppm at any time or 2) an increase in H_2_ ≥ 20 ppm or in CH_4_ ≥ 10 ppm above the baseline between 20 and 90 minutes after lactulose ingestion [19].

### 2.3. Statistical Analysis

We analyzed the data of the 10 UDCA-treated and nine untreated patients, using the Kolmogorov–Smirnov test to determine whether continuous variables (NDI-K, the sum of hydrogen/methane gas values for 90 minutes, blood chemistry) were normally distributed, and *p* > 0.05 was significant. Thus, we compared the continuous variables between groups at baseline and after two months using the independent t-test and a paired t-test for comparison. We also compared categorical variables, including sociodemographic variables, baseline clinical variables, proportion of patients with adequate relief of FD symptoms, and SIBO prevalence in both groups using Fisher’s exact test or the chi-squared test.

## 3. Results

Table 1 shows the baseline characteristics of the subjects; none of the variables (age, weight, NDI-K, alcohol consumption per week and *Helicobacter* infection) showed statistically significant differences between the two groups. We classified subjects with postprandial fullness and early satiation into PDS, and those with epigastric pain and epigastric burning into EPS and the difference in prevalence of PDS and EPS in the two groups was not statistically different. One patient in the UDCA treatment group and three in the untreated group manifested both PDS and EPS.

Figure 2 shows the NDI-K scores at baseline and at the end of the study (after two months), suggesting a statistically significant decrease in NDI-K scores after two months in the group treated with UDCA (baseline vs. 2 months: 37.2 ± 19.5 vs. 20.2 ± 11.6, *p* = 0.022). After two months, the NDI-K scores of the UDCA—and untreated groups also showed statistically significant differences (20.2 ± 11.6 vs. 37.6 ± 21.0, *p* = 0.037). Furthermore, although there was no statistically significant difference, a greater proportion of patients in the UDCA group compared with the untreated group had relief of FD-related symptoms during the study period (60.0% vs. 22.2%, *p* = 0.170; Figure 3).

Table 2 shows the prevalence of SIBO for each group based on hydrogen or methane gas production at baseline and after two months. In the UDCA group, the number of subjects with hydrogen-producing SIBO decreased from 4 (40%) to 1 (10%) after two months, and the number of those with methane-producing SIBO decreased from 8 (80%) to 4 (40%). In the untreated group, the number of patients with SIBO that produced hydrogen at baseline and two months later remained unchanged (2). The number of subjects with methane-producing SIBO decreased from 9 (100%) to 4 (44%). However, there was no statistically significant difference between the two groups at baseline or after two months.

In the UDCA-treated group, four patients had SIBO that produced both hydrogen and methane gas and the three untreated patients who had SIBO produced both hydrogen and methane gas. Two months later, in the UDCA group, there was one patient with SIBO that produced both gases and one in the untreated group (data not shown). Figure 4 shows the sum of the hydrogen/methane breath test values for 90 minutes at baseline and at the end of the study and the sum of methane values over 90 minutes in the UDCA group showed a statistically significant decrease (baseline vs. after 2 months: 68.2 ± 18.3 vs. 44.8 ± 30.6, *p* = 0.026). There were no statistically significant differences between the two groups in laboratory parameters (white blood cell, hemoglobin, alkaline phosphatase, total cholesterol, albumin, fasting glucose, ALT, AST and GGT) monitored for two months in this study (Table 3).

## 4. Discussion

In the present study, we investigated the effects of UDCA in FD patients with SIBO and found that the methane levels over 90 minutes were significantly reduced following FD symptom improvement in the UDCA group. The number of methane-producing SIBO patients decreased from nine to five in the untreated group and from eight to four in the UDCA-treated group. Thus, the UDCA-treated group produced less methane in the LBT than the untreated group and these results contributed to the reduction in FD symptoms, which was consistent with the higher proportion of patients who had adequate relief of FD symptoms in the test group than in the untreated group.

FD is a condition in which symptoms are not related to organic disease. Although the cause is unknown, several studies suggest that SIBO is associated with FD. In one case-control study, of the 23 dyspeptic patients, 13 (56.5%) showed positive results for hydrogen-producing SIBO, while no SIBO was detected in the control group [7]. Tziatzios et al. demonstrated that FD symptoms appear due to gas released by excessive fermentation of carbohydrates ingested and increased proliferation of gut bacteria in the small intestine [20]. A recent study showed that the levels of pro-inflammatory interleukin-1 α and β were higher in the intestinal mucosa in subjects with SIBO than in those without SIBO and that SIBO may trigger abdominal symptoms via intestinal inflammation [21]. In a systematic review and meta-analysis, researchers reported that the pathogenesis of FD is characterized by microinflammation in the form of local immune cell infiltration, particularly in the small intestine [22]. Furthermore, the use of non-absorbing antibiotic, rifaximin, for two weeks in FD patients led to improvements in global dyspeptic symptoms of belching and postprandial fullness/bloating [9]. Taken together, these results suggest that FD results from small intestinal inflammation caused by SIBO.

In humans, UDCA is a secondary bile acid generated by the metabolism of primary bile acid, chenodeoxycholic acid, and exhibits hydrophilic and potentially cytoprotective properties [23]. In many animal studies, the UDCA induced immune suppression, cellular protection, and suppressed inflammation [24,25,26]. In addition, these protective effects of UDCA are not limited to systemic inflammation, because UDCA also suppressed small intestinal inflammation through decreased bacterial translocation, increased mucin production and inhibition of lipopolysaccharide-induced increased intestinal permeability and enterocyte apoptosis in a mouse model [13]. In short, the preclinical studies involving various animal models suggest that UDCA may prevent or treat chronic inflammation of the small intestine such as SIBO. Indeed, our study showed a reduction in methane gas and improvement in FD symptoms in the UDCA-treated group, which may be due to the antimicrobial and anti-inflammatory roles of UDCA.

One of the interesting findings of this study is that patients with FD carried more methane-producing bacteria than hydrogen-producing bacteria, suggesting that FD symptoms may be associated with methane rather than hydrogen. Although there is no evidence explaining the relationship between methane gas and FD, several possibilities exist. Increased methane in the small intestine decreases the intestinal motility, which slows the intestinal transit [27]. At the same time, decreased movement of the small intestine exacerbates gas production via SIBO metabolism of carbohydrates, which can cause a feeling of satiety or early fullness after eating [28]. In addition, methane, a gas produced by SIBO, augments small intestinal contractile activity, causing patients to feel pain [27]. Taken together, increased methane gas in the small intestine and reduced intestinal motility can trigger FD, characterized by postprandial fullness, early satiation and epigastric pain. However, we conducted this study with a small number of subjects, and no large-scale studies have investigated the relationship between FD according to the gas type produced in SIBO patients. Therefore, additional studies are needed in the future to confirm these results.

Our study has several limitations. The first was that we openly labeled our study participants into an untreated group and a UDCA group without a placebo group. Therefore, the placebo effect cannot be ruled out, although an objective breath test may help overcome this limitation. Second, the breath test using lactulose can have a false-positive effect in subjects with rapid bowel movements [29]. However, LBT can be used to diagnose bacterial overgrowth beyond the jejunum. LBT is noninvasive, and is commonly used in clinical practice, as it is recommended by the North American Consensus [19]. Third, this was an exploratory pilot study investigating the effects of UDCA in FD patients, and as such, the number of potential subjects was not sufficient. Thus, we did not use an official sample size and selected a size that was appropriate for addressing our study objectives. However, despite these limitations, our study is the first randomized controlled trial evaluating the effects of UDCA in FD patients with SIBO. Our findings of improved FD symptoms and reduced methane production during the 90 minutes of LBT in the UDCA-treated group compared with the untreated group are valuable.

In summary, the results of the first preliminary randomized controlled human study showed that treatment with UDCA at a dose of 100 mg three times daily for 60 days provides better relief of FD symptoms and reduced methane levels in LBT compared with the untreated group. However, well-designed, large-scale studies are needed to confirm the findings.

## Figures and Tables

**Figure 1 nutrients-12-01410-f001:**
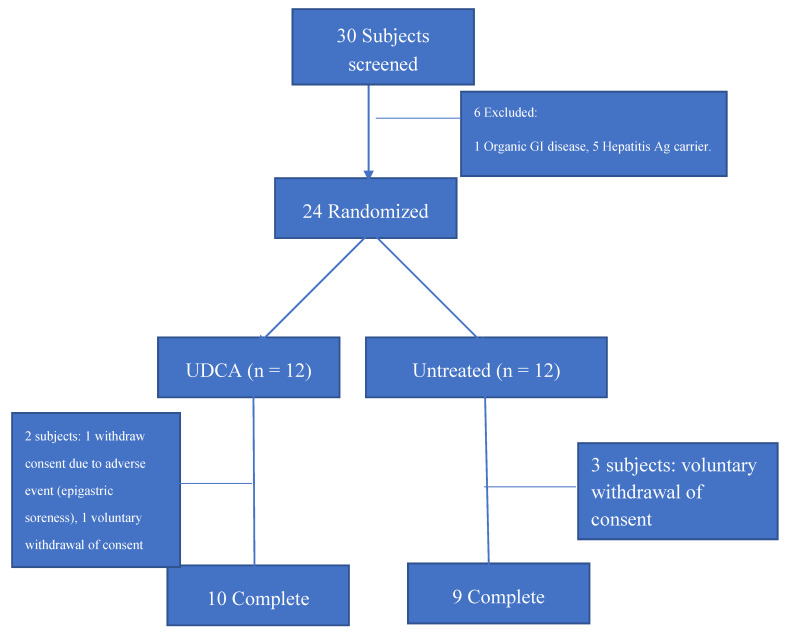
Flow chart outlining the current study protocol.

**Figure 2 nutrients-12-01410-f002:**
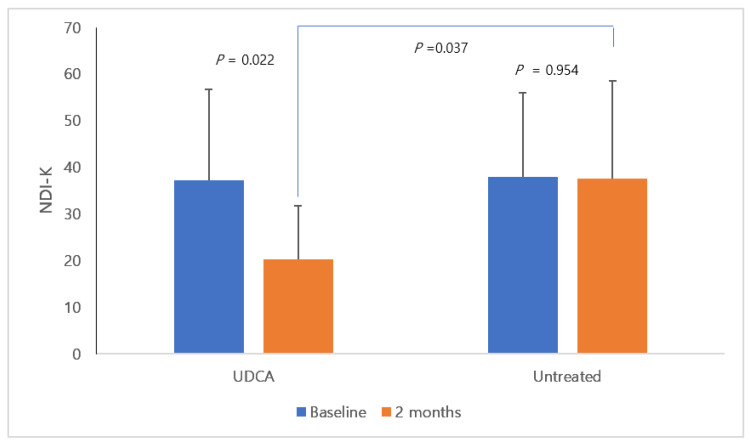
NDI-K scores at baseline and at 2 months. In the group using UDCA, there was a statistically significant decrease in NDI-K score after 2 months (UDCA group: baseline vs. after 2 months; 37.2 ± 19.5 vs. 20.2 ± 11.6, *p* = 0.022; untreated group: 38.0 ± 18.0 vs. 37.6 ± 21.0, *p* = 0.950). In addition, there was a statistically significant difference in the NDI-K score between the UDCA-treated group and the untreated group at 2 months (*p* = 0.037). Bar graph shows the mean with the standard deviation.

**Figure 3 nutrients-12-01410-f003:**
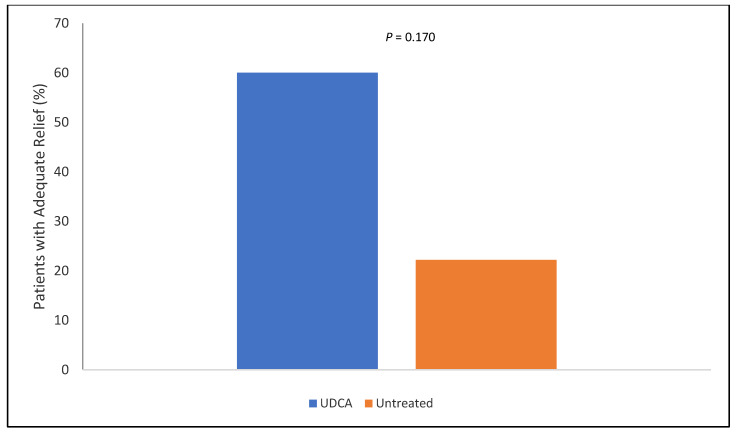
Percentage of patients with adequate relief of FD symptoms at 2 months. Two months later, the number of patients with adequate symptom relief were higher in the UDCA treatment group than in the untreated group, but the difference was not statistically significant (60.0% vs. 22.2%, *p* = 0.170). Adequate relief was defined as NDI-K reduction of more than 30% compared with baseline at 2 months. *p* was calculated with Fisher’s exact test.

**Figure 4 nutrients-12-01410-f004:**
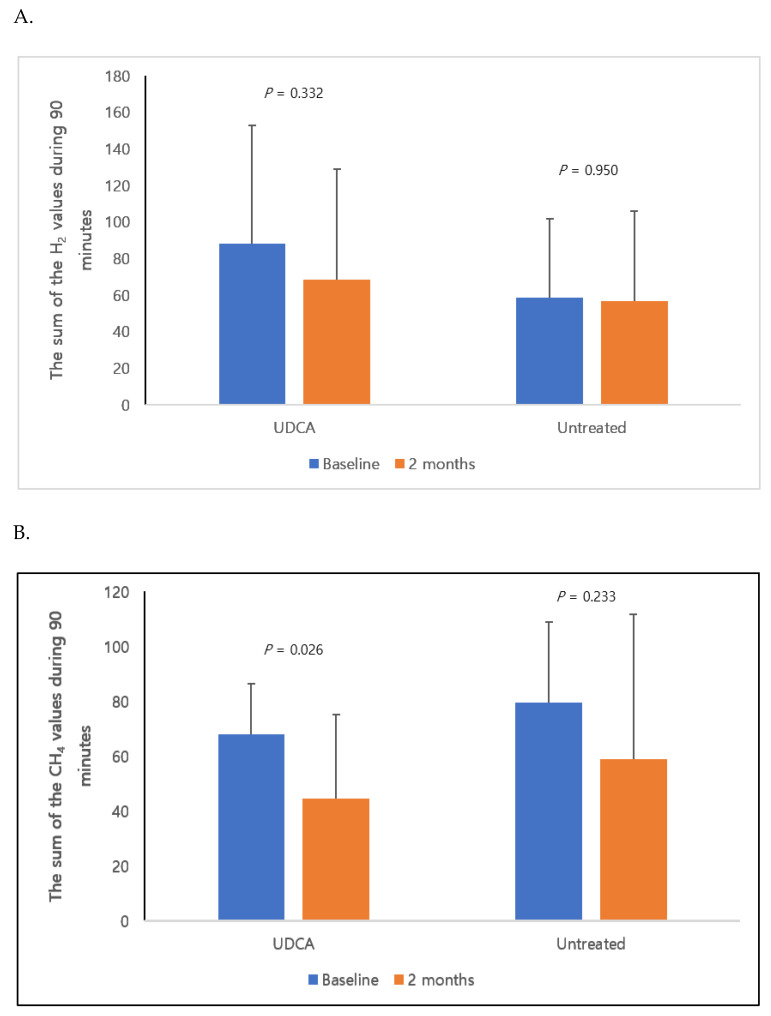
Bar graph showing the sum of hydrogen and methane gases for 90 minutes at baseline and 2 months later in each group. Panel (**A**) shows that in both groups, the sum of hydrogen gas for 90 minutes after 2 months compared with the baseline was not significantly different (UDCA group: 88.0 ± 64.8 vs. 68.4 ± 60.7, *p* = 0.332; untreated group: 58.5 ± 43.6 vs. 56.7 ± 49.3, *p* = 0.950). In contrast, panel (**B**) shows that the sum of methane gases for 90 minutes decreased statistically significantly after 2 months from baseline in the UDCA treatment group (UDCA group: 68.2 ± 18.3 vs. 44.8 ± 30.6, *p* = 0.026; Untreated group: 79.3 ± 60.7 vs. 58.7 ± 52.9, *p* = 0.233). Bar graph shows the mean with the standard deviation. H_2_, hydrogen; CH_4_, methane.

**Table 1 nutrients-12-01410-t001:** Baseline characteristics of the study subjects.

	UDCA (*n* = 10)	Untreated (*n* = 9)	*p*
Age (years)	50.4 ± 9.7	48.3 ± 14.6	0.719
Weight (kg)	69.0 ± 12.7	68.7 ± 14.2	0.972
Men (*n*,%)	6, 60%	5, 55%	0.845
Nepean dyspepsia index-K (score)	37.2 ± 19.5	38.0 ± 18.0	0.927
FD subtype			
PDS (*n*,%)	9, 90%	6, 66%	0.576
EPS (*n*,%)	1, 10%	3, 33%	0.672
*Helicobacter pylori* infection prevalence (*n*,%)	3, 30%	2, 22%	0.708
Current smoker (*n*,%)	0, 0%	1, 11%	0.541
Weekly alcohol intake (g/week)	93.5 ± 84.6	72.0 ± 56.8	0.608
Diabetes (*n*,%)	1, 10%	2, 22%	0.653
Hypertension (*n*,%)	0, 0	1, 11	0.541
Dyslipidemia (*n*,%)	1, 10	1, 11	0.452

UDCA—ursodeoxycholic acid; FD—functional dyspepsia; PDS—post-prandial distress; EPS—epigastric pain syndrome.

**Table 2 nutrients-12-01410-t002:** Results of lactulose breath test at baseline and at 2 months according to hydrogen or methane gas released.

	Positive Hydrogen Gas-Producing SIBO		Positive Methane Gas-Producing SIBO	
	Baseline	2 Months	Baseline	2 Months
UDCA (*n* = 10)	4 (40%)	1 (10%)	8 (80%)	4 (40%)
Untreated (*n* = 9)	2 (22%)	2 (22%)	9 (100%)	4 (44%)
*p*	0.874	0.784	0.510	0.845

SIBO—small intestinal bacterial overgrowth; UDCA—ursodeoxycholic acid. The *p* values of the two groups were calculated by Fisher’s exact test at baseline and at 2 months.

**Table 3 nutrients-12-01410-t003:** Blood chemistry of subjects at baseline and at 2 months.

	Baseline			2 Months		
	UDCA (*n* = 10)	Untreated (*n* = 9)	*p*	UDCA (*n* = 10)	Untreated (*n* = 9)	*p*
White blood cells (x 10^3^)	5.7 ± 1.1	6.2 ± 1.7	0.456	6.6 ± 1.7	5.5 ± 0.7	0.109
Hemoglobin (g/dL)	14.2 ± 2.0	13.4 ± 1.8	0.408	14.4 ± 0.8	14.3 ± 1.6	0.912
Alkaline phosphatase (IU/L)	60.8 ± 12.7	64.4 ± 13.4	0.552	60.6 ± 10.9	66.6 ± 18.7	0.395
Total cholesterol (mg/dL)	192.7 ± 27.6	203.3 ± 23.4	0.381	192.4 ± 41.2	198.3 ± 34.6	0.740
Fasting glucose (mg/dL)	100.3 ± 11.7	98.3 ± 14.0	0.743	99.4 ± 8.5	97.1 ± 15.2	0.698
ALT (IU/L)	28.6 ± 20.4	28.0 ± 19.0	0.948	29.3 ± 21.3	33.7 ± 29.0	0.705
AST (IU/L)	27.6 ± 11.4	26.4 ± 8.9	0.811	27.9 ± 10.4	27.7 ± 11.8	0.981
GGT (IU/L)	57.3 ± 15.3	56.0 ± 60.4	0.864	54.8 ± 12.1	55.0 ± 43.7	0.786

UDCA—ursodeoxycholic acid; ALT—alanine aminotransferase; AST—aspartate aminotransferase; GGT—gamma-glutamyltransferase.

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
