# Peer review of "The Effect of Ursodeoxycholic Acid on Small Intestinal Bacterial Overgrowth in Patients with Functional Dyspepsia: A Pilot Randomized Controlled Trial"

_nutrients, 2020, doi:10.3390/nu12051410_

Round 1
Reviewer 1 Report
Major comments
1) Have the authors excluded patients with IBS from FD? It is generally considered that the predominant methane increase in patients with SIBO is associated with constipation. I wonder if methane-producing patients with SIBO were due to concomitant IBS-C but not simply FD.
2) In figure 3, the authors should not mention the tendency in symptom relief difference between 2 groups as the difference was not statistically significant.
Minor comments
1) For such a small-scale study, the authors should follow strict criteria in diagnosis of SIBO. A positive LBT should be defined as H2 ≥ 20ppm and methane ≥ 10ppm increase within 90min above basal values. Increased basal H2 concentrations should not be considered as positive LBT.
2) How many patients did they have a dual peak profile in H2 and methane during LBT?
3) The authors should be careful about the bold and the font. They should not use the different font without meaning (ex. Bar graph shows the mean with the standard deviation).
Typos
1) In table 1, Men (n, %) = (6. 10). Should read: Men (n, %) = (6. 60).
2) Page 6: he subjects with methane-producing SIBO decreased from 9 (100%) to 4 (44%). Should read: The subjects with methane-producing SIBO decreased from 9 (100%) to 4 (44%).
Author Response
Major comments
1) Have the authors excluded patients with IBS from FD? It is generally considered that the predominant methane increase in patients with SIBO is associated with constipation. I wonder if methane-producing patients with SIBO were due to concomitant IBS-C but not simply FD.
→ Thank you for your deep review. Our study also investigated whether patients with FD have IBS according to ROME IV criteria. No patients complained of abdominal pain corresponding to ROME IV. However, constipation patients according to the Bristol stool scale had 4 out of 24 enrolled in this study. Two of these patients were dropped out, and two were assigned to the UDCA administration group, showing improved symptoms of dyspepsia and constipation. Importantly, the fact that after UDCA treatment, methane gas reduction and FD symptoms improved suggests that methane gas is also involved in constipation, but also related to FD. However, please understand that the descriptions of these contents have few cases and do not correspond to the purpose of this paper, so the authors did not describe them.
2) In figure 3, the authors should not mention the tendency in symptom relief difference between 2 groups as the difference was not statistically significant.
→ We agree with your opinion. The authors revised the description of tendency in Figure 3 as follows; Two months later, there were more patients with adequate symptom relief in the UDCA treatment group than in the non-administered group, but the difference was not statistically significant (60.0% vs. 22.2%, P = 0.170).
Minor comments
1) For such a small-scale study, the authors should follow strict criteria in diagnosis of SIBO. A positive LBT should be defined as H2 ≥ 20ppm and methane ≥ 10ppm increase within 90min above basal values. Increased basal H2 concentrations should not be considered as positive LBT.
→ Thanks for your comments. There is still disagreement among researchers whether the definition of SIBO includes an increase in hydrogen gas measured at baseline, but as recently recommended by the North American Consensus, the authors removed the increase in baseline hydrogen gas values ​​from the definition of SIBO. We also revised the related results and sentences in red.
2) How many patients did they have a dual peak profile in H2 and methane during LBT?
→ In our study data, none of the patients showed double peaks, which may be due to the examination time for 2 hours instead of 3 hours. In our clinical experience, we have not experienced many patients with double peaks within 2 hours.
3) The authors should be careful about the bold and the font. They should not use the different font without meaning (ex. Bar graph shows the mean with the standard deviation).
→ Thanks for your comments. The non-constant font size was matched to a constant size.
Typos
1) In table 1, Men (n, %) = (6. 10). Should read: Men (n, %) = (6. 60).
→ Thank you for your point. We revised the numbers in Table 1 in red.
2) Page 6: he subjects with methane-producing SIBO decreased from 9 (100%) to 4 (44%). Should read: The subjects with methane-producing SIBO decreased from 9 (100%) to 4 (44%).
→ Thank you for your point. We wrote it in a revised sentence in the result section.

Reviewer 2 Report
dear Authors
This research is very important and more evidence on the effect of Ursodeoxychoil Acid in Functional Dyspeptic Patients with Small Intestinal Bacterial Overgrowth is needed; however, the following are my comments:
1- The introduction needs more elaboration and highlights the work conducted by other researchers.
2- The sample size is very small; limit the study impact for generating strog evedence ; therfore I suggest to avaoid using trail; you can call it small scale study. please change the title.
3- otherwise the methadolgy is well written, supported by deyailed data analysis and discussion.
Author Response
Reviewer 2
This research is very important and more evidence on the effect of Ursodeoxychoil Acid in Functional Dyspeptic Patients with Small Intestinal Bacterial Overgrowth is needed; however, the following are my comments:
1- The introduction needs more elaboration and highlights the work conducted by other researchers.
→ We appreciate your feedback and have added some relevant research to describe in the introduction. The added content is shown in red.
2- The sample size is very small; limit the study impact for generating strog evedence ; therfore I suggest to avaoid using trail; you can call it small scale study. please change the title.
→ We agree with your comments and have changed the title to: Effect of Ursodeoxychoil Acid in Functional Dyspeptic Patients with Small Intestinal Bacterial Overgrowth: A Pilot Randomized Controlled Trial
3- otherwise the methadolgy is well written, supported by deyailed data analysis and discussion.

Round 2
Reviewer 1 Report
The revised manuscript is much improved. I am satisfied with the corrections for my comments.